

# A dynamical reconstruction of the Last Glacial Maximum ocean state constrained by global oxygen isotope data

Charlotte Breitkreuz[1], André Paul[1], and Michael Schulz[1]

[1]MARUM - Center for Marine Environmental Sciences and Faculty of Geosciences, University of Bremen, Bremen, Germany

**Correspondence:** Charlotte Breitkreuz (cbreitkreuz@marum.de)

**Abstract.** Combining ocean general circulation models with proxy data via data assimilation is a means to obtain estimates of past ocean states that are consistent with model physics as well as with proxy data. The climate during the Last Glacial Maximum (LGM, 19–23 ka) was substantially different from today. Even though boundary conditions are comparatively well known, the large-scale patterns of the ocean circulation during this time remain uncertain. Previous efforts to combine ocean

models with proxy data have shown dissimilar results regarding the state of the ocean, in particular of the Atlantic Meridional Overturning Circulation. Here, we present a new LGM ocean state estimate that extents previous estimates by using global benthic as well as planktic data on the oxygen isotopic composition of calcite. It is further constrained by global seasonal and annual sea surface temperature (SST) reconstructions. The estimate shows an Atlantic Ocean that is similar to the Late Holocene Atlantic Ocean but with a reduced formation of Antarctic Bottom Water, in contrast to results of previous studies.

The results indicate that SST and oxygen isotopic data alone do not require the presence of a shallower North Atlantic Deep Water and a more extensive Antarctic Bottom Water, and highlight the need for more proxy data of different types to obtain reliable ocean state estimates. Additional adjoint sensitivity experiments reveal that data from the deep North Atlantic and from the global deep Southern Ocean are most important to constrain the Atlantic Meridional Overturning Circulation.

# 1  Introduction

The climate during the Last Glacial Maximum (LGM, 19–23 ka) was substantially different from today. The $CO_2$ concentration in the atmosphere was significantly lower than during the pre-industrial era and large parts of the northern hemisphere were covered by ice sheets resulting in a sea level that was about 130 m lower than today (Mix et al., 2001; Clark et al., 2009). Even though the boundary conditions, such as incoming solar radiation, ice sheet extent, and atmospheric $CO_2$ during the

LGM are comparatively well known, the large-scale patterns of the ocean circulation, for example, the strength of the Atlantic Meridional Overturning Circulation (AMOC) or the extend of the North Atlantic Deep Water (NADW), remain uncertain. The ocean circulation plays a major role in the climate system as it transports and stores massive amounts of energy and nutrients. It is thought to have a large influence on the amount of carbon that is stored in the deep ocean and, therefore, on the carbon





cycle and the atmospheric $CO_2$ concentration on glacial-interglacial timescales (Broecker, 1982; Sigman and Boyle, 2000; Lund et al., 2011).

Many studies indicate the presence of a shallower NADW and a more sluggish AMOC during the LGM compared to today (Lynch-Stieglitz et al., 2007). This hypothesis is supported by studies using different proxies, for example, the carbon isotopic composition of calcite ($\delta^{13}C$, e.g., Duplessy et al., 1988; Sarnthein et al., 1994; Curry and Oppo, 2005), the oxygen isotopic ratio of calcite ($\delta^{18}O_c$, Lund et al., 2011), or sediment pore water measurements (Adkins et al., 2002). The radiogenic isotope ratio of protactinium-231 and thorium-230 ($^{231}Pa/^{230}Th$) can be used to reconstruct flow rates but results are inconclusive about the strength of the AMOC during the LGM (e.g., McManus et al., 2004; Gherardi et al., 2005; Lippold et al., 2016). Model simulations have as well shown dissimilar results for the glacial Atlantic Ocean. The PMIP2 simulations showed different AMOC responses to glacial forcing (e.g. Otto-Bliesner et al., 2007; Weber et al., 2007), whereas the latest PMIP3 simulations found a stronger and deeper AMOC (Muglia and Schmittner, 2015). Most recently Muglia et al. (2018) found that a simulated weaker AMOC resulted in the best fit between a physical-biogeochemical ocean model and global $\delta^{13}C$, radiocarbon, and $\delta^{15}N$ data.

Combining climate models and proxy data via data assimilation is a means to obtain more reliable simulations of past ocean states. The adjoint method (Wunsch, 1996; Errico, 1997) yields a dynamical reconstruction of the ocean state, that is, an optimized model simulation that represents the best fit of the model and proxy data in a least squares sense. Several dynamical reconstructions of the LGM ocean state obtained with general circulation models exist (Winguth et al., 2000; Dail and Wunsch, 2014; Kurahashi-Nakamura et al., 2017; Amrhein et al., 2018). The estimates are partly only based on surface proxy data and focus on reconstructing the upper ocean (Dail and Wunsch, 2014; Amrhein et al., 2018) and partly include the assimilation of deep-ocean data (Winguth et al., 2000; Kurahashi-Nakamura et al., 2017), which is important for constraining the deep-ocean circulation and water masses. The longest adjoint LGM ocean state estimate (Kurahashi-Nakamura et al., 2017) is 400 years long and constrained by global sea surface temperature (SST) and benthic $\delta^{18}O_c$ and $\delta^{13}C$ data from the Atlantic Ocean. The surface levels of the model domain were excluded for $\delta^{18}O_c$ and $\delta^{13}C$ because the processes that are prevalent for these tracers in the upper layers were not included in their model. Kurahashi-Nakamura et al. (2017) found a stronger, but shallower AMOC compared to the modern ocean.

The use of benthic as well as of planktic data may be especially important because the isotopic signature of deep water masses is defined at the surface where deep water is formed, which in turn can be constrained by planktic data. Additionally, the use of deep-ocean data from the global, not only the Atlantic Ocean, might be important to constrain the global overturning circulation (Breitkreuz et al., 2019, in review). A recent LGM ocean state estimate was obtained from a Kalman smoother method in combination with a state reduction approach (Breitkreuz et al., 2019, in review). Their estimate covers 2,000 model years and uses benthic as well as planktic data on the oxygen isotopic composition of calcite from the global ocean. They obtained a shallower and weaker AMOC compared to the modern ocean, however, a substantial model-data misfit remained.

In this study we present a new 400-year LGM ocean state estimate obtained from the adjoint method that extends the estimate of Kurahashi-Nakamura et al. (2017) by using global oxygen isotopic data from benthic as well as from planktic foraminifera, and global seasonal and annual SST reconstructions (MARGO Project Members, 2009), however, without benthic $\delta^{13}C$ data.



We use a general circulation model enhanced with a water isotope module (Völpel et al., 2017, 2018) that enables it to simulate the oxygen isotopic composition of seawater ($\delta^{18}O_{sw}$) in the complete water column. This study investigates the robustness of previous ocean state estimates and the influence of the additional data constraint placed by global surface data and deep-ocean data from outside of the Atlantic Ocean on the estimate, in particular, on the AMOC strength and the vertical extent of the

NADW. We additionally perform adjoint sensitivity experiments based on the 400-year estimate that provide the sensitivity of the AMOC with respect to the global ocean temperature and salinity and, therefore, indicate which areas of the global ocean are most important for constraining the AMOC.

## 2  Material and Methods

### 2.1  Model

We used the coupled ocean-sea-ice Massachussets Institute of Technology general circulation model (MITgcm; Marshall et al., 1997; MITgcm Group, 2016) in the same global configuration as used by Breitkreuz et al. (2018). The configuration uses a cubed-sphere grid (Ronchi et al., 1996) with a horizontal resolution of about 2.8° and 15 vertical levels. Outgoing radiation, wind stress, and evaporation were computed by bulk formulae (Large and Yeager, 2004) and a GM/Redi scheme (Redi, 1982; Gent and Mcwilliams, 1990) was used to parameterize subgrid-scale mixing. In this study the model was driven by monthly air temperature, meridional and zonal wind velocities, wind speed, specific humidity, precipitation, downward shortwave radiation, downward longwave flux, and river run-off, based on a fully coupled LGM simulation of the Community Climate System Model Version 3 (Merkel et al., 2010). Following Völpel et al. (2018) and Breitkreuz et al. (2019, in review), the Mediterranean Sea was excluded from the model domain, because a shallow passage trough the Strait of Gibraltar was not possible due to the coarse vertical resolution. The isotopic composition of seawater

$$\delta^{18}O_{sw} = \left( \frac{^{18}O/^{16}O}{R_{VSMOW}} - 1 \right) \cdot 1,000\%o$$

with respect to the Vienna Standard Mean Ocean Water (VSMOW, $R_{VSMOV} = 2,005.2 \cdot 10^{-6}$, Gonfiantini, 1978) can be computed from the concentration of the isotopes $H_2^{16}O$ and $H_2^{18}O$, which are included in the model by a water isotope module (Völpel et al., 2017). The climatological isotopic composition of precipitation and water vapor need to be prescribed and were obtained from a water isotope-enabled LGM simulation with the Community Atmosphere Model version 3.0 (IsoCAM3.0, Tharammal et al., 2013). To apply the adjoint method, the adjoint of the model code needs to be obtained. The MITgcm is

tailored to automatic differentiation and the adjoint code can be generated with a source-to-source translator (Giering and Kaminski, 1998; Giering, 2000).

### 2.2  Proxy Data and Uncertainties

The LGM estimate presented in this study is constrained by seasonal and annual SST LGM-Late Holocene (LH) anomaly reconstructions and by a global compilation of data on the oxygen isotopic composition of foraminiferal calcite ($\delta^{18}O_c$) LGM-

LH anomaly. The oxygen isotopic composition of seawater ($\delta^{18}O_{sw}$) is a passive tracer of water masses in the deep ocean



and water masses have distinct $\delta^{18}O_{sw}$ signals (Breitkreuz et al., 2018). The oxygen isotopic composition of foraminiferal calcite preserves the $\delta^{18}O_{sw}$ and temperature signal of the past ocean. With the use of anomalies, we aim at eliminating species-specific vital effects.

We used the SST anomaly reconstructions by the MARGO Project Members (2009), which are based on a combination of multiple microfossil-based and geochemical proxies. In this study we used the annual as well as the seasonal temperature estimates for July, August, September (JAS) and January, February, March (JFM). The MARGO Project Members (2009) additionally provide estimates of the respective uncertainties of the reconstructed temperature and a mean reliability index. Following Breitkreuz et al. (2019, in review), we averaged the raw data and their respective uncertainties onto our model grid according to the reliability index as proposed by the MARGO Project Members (2009). In total, 1,120 grid cells were filled with SST data including the seasonal and the annual data.

We used the $\delta^{18}O_c$ compilation including uncertainties used by Breitkreuz et al. (2019, in review), which include global $\delta^{18}O_c$ data from planktic as well as from benthic species from different sources. An overview over the $\delta^{18}O_c$ data sets they used can be found in Table 1. Breitkreuz et al. (2019, in review) used the same model grid and averaged the data and their uncertainties onto the model grid. The planktic $\delta^{18}O_c$ data were assigned to the first depth level of the model covering 0–50 m. The depth levels for the benthic data were determined by subtracting 130 m from the core depth to account for the mean sea level change during the LGM (Clark et al., 2009). We used their averaged data set containing 200 grid cells with benthic and 136 grid cells with planktic data. Figure 1 shows the locations of the combined LGM SST and $\delta^{18}O_c$ data from benthic and planktic foraminifera.

**Table 1.** Data sets utilized to constrain the LGM estimate. The $\delta^{18}O_c$ data sets were previously used by Breitkreuz et al. (2019, in review). Number of data points refers to the available LGM-LH anomaly data points in the original data sets. JFM = January, February, March. JAS = July, August, September.

| Data type (LGM-LH anomalies) | Area | # Data points | Reference |
| --- | --- | --- | --- |
| Annual SST | Global Ocean | 667 | MARGO Project Members (2009) |
| JFM SST | Global Ocean | 638 | MARGO Project Members (2009) |
| JAS SST | Global Ocean | 518 | MARGO Project Members (2009) |
| Planktic $\delta^{18}O_c$ | Global Ocean | 136 | Waelbroeck et al. (2014) |
| | Global Ocean | 114 | Caley et al. (2014) |
| | Global Ocean | 123 | Breitkreuz et al. (2019, in review; LGM) and Waelbroeck et al. (2005, LH) |
| Benthic $\delta^{18}O_c$ | Atlantic Ocean | 163 | Marchal and Curry (2008) |
| | Global Ocean | 114 | Caley et al. (2014) |
| | Atlantic Ocean | 5 | Völpel et al. (2018) |





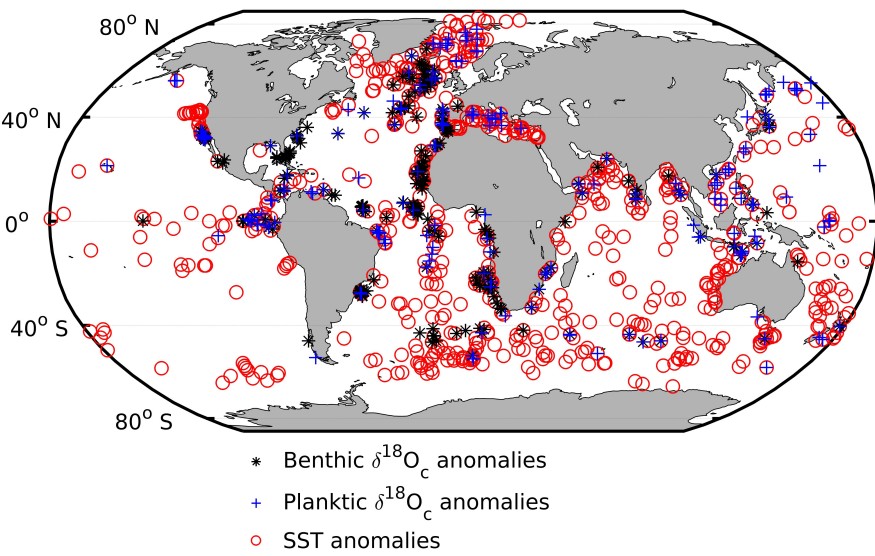

**Figure 1.** Locations of proxy data used in this study: $\delta^{18}O_c$ data from planktic and benthic foraminifera and annual, JFM, and JAS SST.

## 2.3 Optimization

We used the adjoint method (Wunsch, 1996; Errico, 1997), also called 4-dimensional variational method (4D-Var), or method of Lagrange multipliers, to obtain an estimate of the LGM ocean that is consistent with the observational data in a least-squares sense as well as with the model physics. The adjoint method minimizes a cost function measuring the model-data misfit by adjusting defined control variables. The control variables can include, for example, initial or boundary conditions, or internal model parameters. Iteratively, the gradient of the cost function close to the current best guess of the control variables is computed from the adjoint model code and used by a quasi-Newton descent algorithm (Nocedal, 1980; Gilbert and Lemaréchal, 1989) to compute adjustments to the control variables.

To create initial conditions for the optimization, the model was first spun-up for 3,000-years using the the original first-guess control variables without the two isotopic tracers and subsequently for another 3,000 years with the isotopic tracers. The adjoint method is typically used to optimize simulations covering 10–50 years (e.g., Köhl et al., 2007; Köhl and Stammer, 2008; Forget et al., 2015) and the longer the simulation, the more difficult it is to achieve a sufficient reduction of the cost function. This might be due to the non-linearity of the model and the increasingly non-convex shape of the cost function with increasing simulation length (Evensen, 2009). To obtain a comparatively long optimized simulation of 400 years, we used a *carry-over* technique (Dail, 2012; Kurahashi-Nakamura et al., 2017; Breitkreuz et al., 2018), where first, a short simulation is optimized and, subsequently, the optimized control variables are used as a first guess for the optimization of a longer run, eventually reaching the desired length. Following Kurahashi-Nakamura et al. (2017) and Breitkreuz et al. (2018), we chose the atmospheric forcing fields (i.e., air temperature, specific humidity, precipitation, zonal and meridional wind velocities, downward shortwave radiation, downward longwave flux, isotopic composition in precipitation and water vapor), the initial





conditions for the physical tracers (salinity and temperature), and the spatially-varying vertical diffusivity as control variables. The optimization with the carry-over method was started with a 50-year long run followed by a 100-, a 150-, a 200-, and finally a 400-year run. We were not able to obtain a significant reduction of the SST model-data misfit in the first 50-year optimization while using the isotopic control variables. This might be due to the non-convex shape of the cost function and the

possibility that the optimization came to a halt in a local minimum. We, therefore, included the isotopic control variables only in the 100-year optimization. Following the 100-year optimization, we excluded the isotopic control variables once again as we already found comparatively big local changes in those control variables.

Following Breitkreuz et al. (2018), the cost function consists of three parts $J = J_{\mathrm{misfit}} + J_{\mathrm{ctrl}} + J_{\mathrm{eq}}$, which quantify the model-data misfit ($J_{\mathrm{misfit}}$), the deviation from the first guess of the control variables ($J_{\mathrm{ctrl}}$), and the model drift ($J_{\mathrm{eq}}$). The first term is

given by

$$
\begin{aligned}
J_{\mathrm{misfit}} = {} & \left( \boldsymbol{SST}^{\mathbf{sim}} - \boldsymbol{SST}^{\mathbf{obs}} \right)^{\top} \mathbf{W}_{\boldsymbol{SST}} \left( \boldsymbol{SST}^{\mathbf{sim}} - \boldsymbol{SST}^{\mathbf{obs}} \right) \\
& + \left( \boldsymbol{\delta^{18}O}_{\mathbf{c}}^{\mathbf{sim}} - \boldsymbol{\delta^{18}O}_{\mathbf{c}}^{\mathbf{obs}} \right)^{\top} \mathbf{W}_{\boldsymbol{\delta^{18}O_c}} \left( \boldsymbol{\delta^{18}O}_{\mathbf{c}}^{\mathbf{sim}} - \boldsymbol{\delta^{18}O}_{\mathbf{c}}^{\mathbf{obs}} \right).
\end{aligned}
$$

Here, $\boldsymbol{SST}^{\mathbf{sim}}$, $\boldsymbol{SST}^{\mathbf{obs}}$, $\boldsymbol{\delta^{18}O}_{\mathbf{c}}^{\mathbf{sim}}$, and $\boldsymbol{\delta^{18}O}_{\mathbf{c}}^{\mathbf{obs}}$ are vectors containing the $\delta^{18}O_c$ and SST anomalies, simulated and observed values, respectively. The vectors of SST's contain annual mean values as well as mean values for January, February,

March and July, August, September. The vectors for $\delta^{18}O_c$ are annual mean values and include values for planktic and benthic data. The simulated values are long-term means for the respective time of the year at the observed grid cells. The long-term mean is computed for a certain time interval at the end of each iteration depending on the respective step in the carry-over process (Table 2). The weighting matrices $\mathbf{W}_{\boldsymbol{\delta^{18}O_c}/\boldsymbol{SST}}$ are the inverse of the error-covariance matrices of the respective proxy data. The uncertainties of the proxy data are assumed to be spatially uncorrelated such that the matrices are diagonal.

The simulated $\delta^{18}O_c$ values were computed from simulated $\delta^{18}O_{sw}$ and temperature $T$ according to the paleo-temperature equation by Shackleton (1974)

$$
T = 16.9 - 4.38 \left( \delta^{18}O_c - \delta^{18}O_{sw} \right) + 0.10 \left( \delta^{18}O_c - \delta^{18}O_{sw} \right)^2.
$$

Beforehand, an offset of 1.1‰ was added to simulated $\delta^{18}O_{sw}$ to account for the global mean change during the LGM (Duplessy et al., 2002) and the values were transferred from the VSMOW to the Vienna Peedee belemnite (VPDB) standard by

subtracting 0.27‰ (Hut, 1987). The simulated LGM-LH anomalies were computed using a state estimate of the modern ocean (Breitkreuz et al., 2018) that was obtained by fitting the MITgcm in the same configuration to modern climatological salinity, temperature and $\delta^{18}O_{sw}$ data with the adjoint method. The use of anomalies enabled us to use one paleo-temperature equation for all species because species-specific vital effects cancel out. Additionally, systematic model errors in the simulated LH and LGM state are assumed to largely neutralize each other.



**Table 2.** Overview over the carry-over process including the lengths of the cost function intervals, the normalized model-data misfit ($J'_{\mathrm{misfit}}$), the normalized deviation from the first-guess control variables ($J'_{\mathrm{ctrl}}$), and the AMOC strength. The normalized values refer to the model-data misfit/deviation from the first-guess control variables divided by the respective number of model-data comparisons/control variables. For $J'_{\mathrm{misfit}}$ respective values for SST and $\delta^{18}O_c$ are given. According to the theory of a $\chi^2$-test, a value of one indicates agreement of the model with the proxy data within their respective uncertainties or an adjustment of the control variables within the assumed uncertainties. Note that $J'_{\mathrm{ctrl}}$ is 0 at the beginning of the optimization as the control variables were not yet modified. Bold numbers highlight the results in the final state estimate.

| | | $J'_{\mathrm{misfit}}$ | | | | |
|---|---|---|---|---|---|---|
| Run length | Cost fun. interval | SST | Plank. $\delta^{18}O_c$ | Bent. $\delta^{18}O_c$ | $J'_{\mathrm{ctrl}}$ | AMOC |
| First guess spin-up (3000 years) | 100 years | 1.7 | 41.7 | 3.9 | 0 | 17.3 Sv |
| 50 years | 25 years | 0.7 | 3.3 | 1.4 | 0.6 | 11.8 Sv |
| 100 years | 60 years | 0.8 | 2.0 | 1.9 | 0.5 | 11.0 Sv |
| 150 years | 60 years | 0.7 | 1.6 | 2.3 | 0.6 | 21.3 Sv |
| 200 years | 60 years | 0.7 | 1.8 | 1.8 | 0.6 | 16.5 Sv |
| **400 years** | **60 years** | **0.7** | **2.1** | **1.9** | **0.6** | **16.1 Sv** |

The second part of the cost function has the form

$$
\begin{aligned}
J_{\mathrm{ctrl}} = \frac{N_{\mathrm{data}}}{N_{\mathrm{ctrl}}} \cdot \Bigg[ & \left( \boldsymbol{T_0}^{\mathrm{orig}} - \boldsymbol{T_0}^{\mathrm{adj}} \right)^{\top} \mathbf{W}_{T_0} \left( \boldsymbol{T_0}^{\mathrm{orig}} - \boldsymbol{T_0}^{\mathrm{adj}} \right) \\
& + \left( \boldsymbol{S_0}^{\mathrm{orig}} - \boldsymbol{S_0}^{\mathrm{adj}} \right)^{\top} \mathbf{W}_{S_0} \left( \boldsymbol{S_0}^{\mathrm{orig}} - \boldsymbol{S_0}^{\mathrm{adj}} \right) \\
& + \sum_{i=1,9} \left( \boldsymbol{F_i}^{\mathrm{orig}} - \boldsymbol{F_i}^{\mathrm{adj}} \right)^{\top} \mathbf{W}_{F_i} \left( \boldsymbol{F_i}^{\mathrm{orig}} - \boldsymbol{F_i}^{\mathrm{adj}} \right) \\
& + \left( \boldsymbol{K}^{\mathrm{orig}} - \boldsymbol{K}^{\mathrm{adj}} \right)^{\top} \mathbf{W}_{K} \left( \boldsymbol{K}^{\mathrm{orig}} - \boldsymbol{K}^{\mathrm{adj}} \right) \Bigg].
\end{aligned}
$$

It measures the deviation of the adjusted (adj) from the original (orig) control variables weighted by the inverse of the respective error covariance matrices $\mathbf{W}$. The control variables and their prior uncertainties are given in Table 3. The value of this part of the cost function increases during the optimization with growing adjustments to the control variables and, therefore, limits the deviation from the first-guess control variables. This term serves as a regularization of the problem as the number of control

10 variables is substantially bigger than the number of observations and the problem would be highly under-determined without this term. As proposed by Kurahashi-Nakamura et al. (2017), the factor $N_{\mathrm{data}} \cdot N_{\mathrm{ctrl}}^{-1}$ is included to balance the contributions of $J_{\mathrm{misfit}}$ and $J_{\mathrm{ctrl}}$. The number of control variables is significantly higher than the number of model-data comparisons and a reduction of $J_{\mathrm{misfit}}$ would be prohibited by the penalty terms without the balancing factor.





**Table 3.** Control variables, assumed prior uncertainties (one standard deviation, $\sigma$), and normalized deviation from the first-guess values after the final phase of the carry-over process ($J'_{\text{ctrl}}$).

| Symbol | Control variable | Unit | $\sigma$ | $J'_{\text{ctrl}}$ |
|--------|------------------|------|----------|--------------------|
| $T_0$ | Initial temperature | °C | $1.00 \cdot 10^0$ | $8.9 \cdot 10^{-1}$ |
| $S_0$ | Initial salinity | - | $1.00 \cdot 10^{-1}$ | $1.5 \cdot 10^0$ |
| $F_1$ | Surface (2-m) air temperature | K | $3.16 \cdot 10^0$ | $1.0 \cdot 10^0$ |
| $F_2$ | Surface (2-m) specific humidity | kg/kg | $1.00 \cdot 10^{-3}$ | $4.7 \cdot 10^{-1}$ |
| $F_3$ | Precipitation | m/s | $3.16 \cdot 10^{-8}$ | $2.1 \cdot 10^{-2}$ |
| $F_4$ | Surf. (10-m) zonal wind vel. | m/s | $3.16 \cdot 10^0$ | $6.1 \cdot 10^{-1}$ |
| $F_5$ | Surf. (10-m) meridional wind vel. | m/s | $3.16 \cdot 10^0$ | $6.8 \cdot 10^{-1}$ |
| $F_6$ | Downward shortwave radiation | W/m$^2$ | $1.00 \cdot 10^1$ | $7.1 \cdot 10^{-2}$ |
| $F_7$ | Downward longwave flux | W/m$^2$ | $1.00 \cdot 10^1$ | $8.3 \cdot 10^{-2}$ |
| $F_8$ | Isotopic ratio of precipitation | - | $1.00 \cdot 10^{-3}$ | $1.1 \cdot 10^{-5}$ |
| $F_9$ | Isotopic ratio of water vapor | - | $1.00 \cdot 10^{-3}$ | $9.3 \cdot 10^{-6}$ |
| $K$ | Vertical diffusion coefficient | m$^2$/s | $1.00 \cdot 10^{-5}$ | $6.8 \cdot 10^{-1}$ |

Following Breitkreuz et al. (2018), the third term of the cost function is given by

$$J_{\text{eq}} = w_{\overline{T}} \sum_{t=1}^{t_{\text{end}}-1} \left( \overline{T}_t - \overline{T}_{t+1} \right)^2 + w_{\overline{\eta}} \sum_{t=1}^{t_{\text{end}}-1} \left( \overline{\eta}_t - \overline{\eta}_{t+1} \right)^2 + w_{\overline{\text{AMOC}}} \sum_{t=1}^{t_{\text{end}}-1} \left( \overline{\text{AMOC}}_t - \overline{\text{AMOC}}_{t+1} \right)^2, \tag{1}$$

where $\overline{T}$ is the annual mean temperature, $\overline{\eta}$ the annual mean global sea surface elevation, and $\overline{\text{AMOC}}$ the annual mean strength of the southward transport at 45° N in the Atlantic Ocean measured at each year $1, .., t_{\text{end}}$ in the cost function interval. This term penalizes the drift in these three quantities and supports that the obtained state estimate is stable and equilibrated. Following Breitkreuz et al. (2018), we determined the weighting factors $w_{\overline{T}, \, \overline{\eta}, \, \overline{\text{AMOC}}}$ (Table 4) empirically such that a sufficient reduction of the cost function was still possible.

**Table 4.** Variables and respective weights ($w$) in the cost function term $J_{\text{eq}}$.

| Symbol | Variable | $w$ |
|--------|----------|-----|
| $\overline{T}$ | Annual mean temperature | $10^6$ |
| $\overline{\eta}$ | Annual mean sea surface elevation | $10^6$ |
| $\overline{\text{AMOC}}$ | Annual mean strength of southward transport at 45° N in the Atlantic Ocean | $10^5$ |

Following Kurahashi-Nakamura et al. (2017) and Breitkreuz et al. (2018), we additionally applied a pre-conditioning of the optimization problem through a normalization of the control variables with factors according to their characteristic size, a





9-point spatial smoothing to the adjustments of the control variables, and set minimum and maximum values for precipitation, specific humidity, downward shortwave radiation, air temperature, and the isotopic ratios in precipitation and water vapor (Table 5). The following results refer to the mean over the last 100 years of the final 400-year optimization, which will be termed LGM400.

**Table 5.** Imposed minimum and maximum values for the control variables following Breitkreuz et al. (2018).

| Variable | Unit | Minimum | Maximum |
|---|---|---|---|
| Precipitation | m/s | 0 | $2.0 \cdot 10^{-6}$ |
| Specific humidity | kg/kg | 0 | $1.0 \cdot 10^{-1}$ |
| Downward shortwave radiation | W/m$^2$ | 0 | 600 |
| Air temperature | K | 183 | 343 |
| Isotopic ratio in precipitation | - | $1.92 \cdot 10^{-3}$ | $2.02 \cdot 10^{-3}$ |
| Isotopic ratio in water vapor | - | $2.13 \cdot 10^{-3}$ | $2.215 \cdot 10^{-3}$ |

## 2.4 Dye-Tracer Experiments

To get a clear visualization of the northern- and southern-source water masses in the estimated LGM Atlantic Ocean, we performed two experiments with passive dye tracers following Gebbie (2014), Kurahashi-Nakamura et al. (2017) and Völpel et al. (2018). In the first experiment, the concentration of the passive tracer was fixed to one in the North Atlantic at the surface between 40° N and 80° N. In the second experiment, the concentration was set to one between 50° S and 60° S below 2,250 m depth. To distribute the dye tracer, the model was run for 2,000 years by repeating the last 100 years of the final 400-year LGM estimate 20 times. For comparison the same experiments were performed with the 400-year LH estimate of Breitkreuz et al. (2018).

## 2.5 Adjoint Sensitivity Experiments

To investigate the sensitivity of the AMOC to data constraints in different regions of the global ocean, we performed the following adjoint sensitivity experiments. An adjoint simulation with the MITgcm provides the sensitivity of the cost function with respect to all state variables. We implemented a cost function that measures the 10-year mean strength of the AMOC streamfunction at 45° N to investigate the sensitivity of the AMOC strength to the spatially-varying ocean state. The sensitivity of the AMOC with respect to the ocean state is in turn an indication of how sensitive the AMOC strength is with regard to data constraints in certain regions of the global ocean. We re-ran the 400-year optimized simulation and its adjoint simulation with the AMOC cost function to obtain the sensitivities.





# 3 Results

## 3.1 Optimization

The optimization with the adjoint method greatly reduced the model-data misfit as measured by the cost function (Table 2). The first guess simulation showed a good agreement with the SST data, but a very large misfit for the planktic and benthic $\delta^{18}O_c$

data. During the carry-over process the model-data misfit for the SST, planktic, and benthic data were reduced by 58.8 %, 95.0 %, and 51.3 %, respectively (Table 2). The normalized model-data misfit ($J'_{\mathrm{misfit}}$) for the SST has a value below one, which indicates that the optimized simulated SST field agrees on average with the data within their uncertainties. Only single data points show a mismatch in areas where other data points indicate agreement (Fig. 2). The simulated first guess surface $\delta^{18}O_c$ showed a high model-data misfit to the planktic $\delta^{18}O_c$ data in the low latitudes and in the western North Atlantic.

The optimization greatly improves the simulated surface $\delta^{18}O_c$ but some small discrepancies remain, mainly in the North Atlantic (Fig. 3). The simulated deep-ocean $\delta^{18}O_c$ shows an overall agreement with the benthic $\delta^{18}O_c$ data but again some discrepancies remain after the optimization, mostly at shallower depth above 2,000 m (Fig. 4).

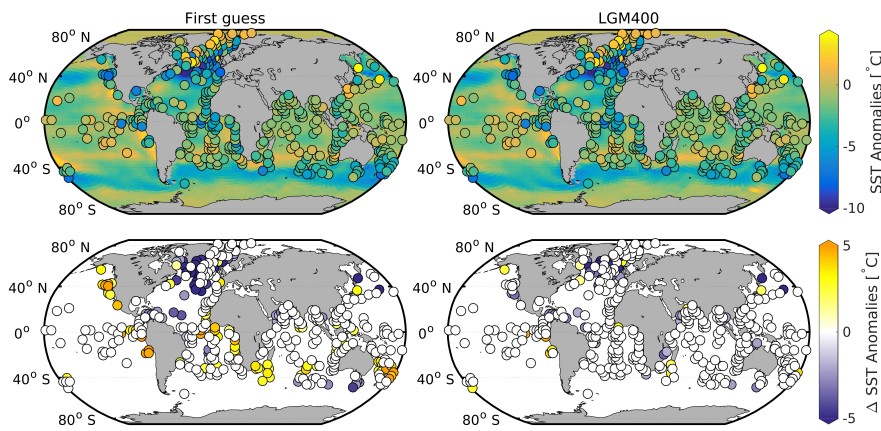

**Figure 2.** Simulated 100-year mean surface field (0–50 m) of SST LGM-LH anomalies in the LGM400 optimization and assimilated annual SST anomaly data (upper panels), and respective model-data differences (lower panels). Differences below the uncertainty of the data are displayed in white.

A further reduction of the cost was either not possible or only a comparatively small reduction of the cost was obtained by locally big, implausible changes in the control variables. Additionally, a further increase of the cost function would likely cause

an overfitting of the model to the SST data for which the normalized cost function shows values lower than one, indicating a model-data misfit smaller than the uncertainties of the data.

As the control variables were adjusted during the optimization, the deviation of the control variables from their first guess ($J_{\mathrm{ctrl}}$) was zero before the optimization and increased during the carry-over process (Table 2). The normalized value ($J'_{\mathrm{ctrl}}$) stayed well below one, which indicates an average change in the control variables within their assumed uncertainties. Except



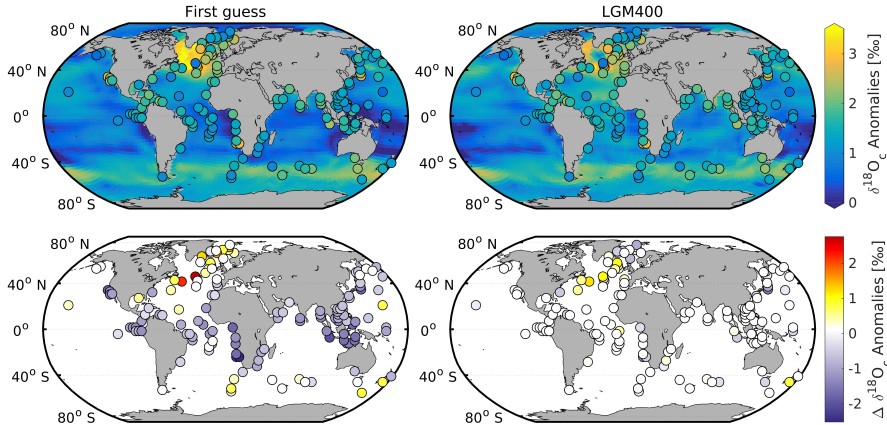

**Figure 3.** Simulated 100-year mean surface field (0–50 m) of $\delta^{18}O_c$ LGM-LH anomalies in the first guess and LGM400 optimization and assimilated planktic $\delta^{18}O_c$ data (upper panels), and respective model-data differences (lower panels). Differences below the uncertainty of the data are displayed in white.

for the initial salinity, the global mean changes in the individual control variable fields stayed within the assumed standard deviations (Table 3). Note that $J'_{\mathrm{ctrl}}$ is a global measure and does not ensure that the adjustments of the control variables are plausible locally.

The drift of the global mean sea surface elevation and the global mean potential temperature are 1.4 cm and 0.035 °C in the last 100 years of the final 400-year optimization. The 100-year mean maximum southward transport in the center of the AMOC is 16.1 Sv with a strengthening of 0.1 Sv in the last 100 years of the 400-year optimization. The southward transport reaches a depth of approximately 3,100 m at 30° S (Fig. 5). During the carry-over process the estimated AMOC strength varies between 11.0 Sv and 21.3 Sv (Table 2).

## 3.2 LGM Ocean State

The estimated LGM surface ocean shows an overall cooling compared to the LH (Fig. 2). The global mean cooling in our estimate is 1.9 °C in the surface layer of the model (0–50 m) and 1.3 °C in the total ocean. The cooling is most pronounced in the North Atlantic. The strong east-west gradient in the North Atlantic as shown by the MARGO data (MARGO Project Members, 2009) is not found in our estimate.

The estimated surface $\delta^{18}O_c$ shows overall higher values compared to the LH, reflecting the lower temperatures. Note that the $\delta^{18}O_{\mathrm{sw}}$ values used to calculate $\delta^{18}O_c$ were corrected for the global ice sheet contribution such that it does not contribute to the $\delta^{18}O_c$ anomalies. The highest values in $\delta^{18}O_c$ anomalies are found in the North Atlantic reflecting high LGM $\delta^{18}O_c$ originating from cold temperatures in this area.

A transect of simulated $\delta^{18}O_{\mathrm{sw}}$ in LGM400 through the Atlantic Ocean shows an isotopically enriched northern-source water mass with its center at about 2,000 m depth similar to the modern NADW and a less enriched intermediate southern-source





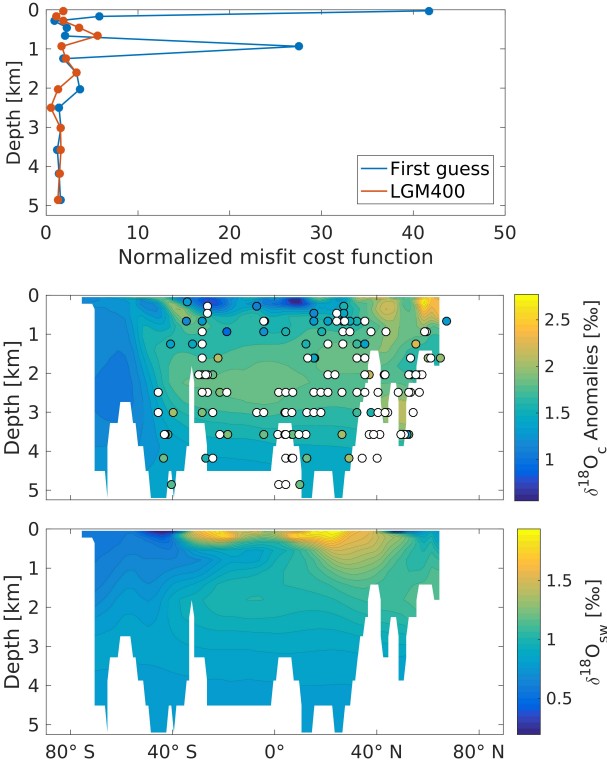

**Figure 4.** Normalized model-data misfit cost function $f'_{\mathrm{misfit}}$ for planktic and benthic $\delta^{18}O_c$ data per depth level (upper panel). Simulated 100-year mean $\delta^{18}O_c$ LGM-LH anomalies (middle panel) and simulated 100-year mean $\delta^{18}O_{sw}$ at a vertical transect through the Atlantic Ocean at 32.5° W in the optimized LGM state and assimilated benthic $\delta^{18}O_c$ data. Data points where the model agrees with the data within their respective uncertainties are displayed in white.

water mass at a depth of about 1,000 m similar to the modern Antarctic Intermediate Water (AAIW, Fig. 4). The influence of a less enriched southern-source deep water mass similar to the modern Antarctic Bottom Water (AABW) is not visible in the simulated optimized $\delta^{18}O_{sw}$. Temperatures in the deep Atlantic Ocean are slightly cooler than during the LH (Fig. 6) and reach a minimum of -1.6 °C in the deep Atlantic Ocean.

5     The dye tracer experiments provide a similar picture of the water masses in the Atlantic Ocean in LGM400 (Fig. 7). In both the LGM and the LH estimate the center of the northern-source NADW is at a depth of about 2,200 m, but in the LGM estimate the NADW penetrates much deeper and further south than during the LH. Accordingly, the southern-source waters reach much further north in the LH estimate.

    The estimated AMOC is only slightly weaker compared to the estimated LH AMOC, and the southward transport reaches a
10  similar depth in both estimates (Fig. 5). The deep counter-clockwise turning cell of the AMOC is stronger in the LH reaching a maximum of 2.1 Sv compared to 0.9 Sv in the LGM estimate. To summarize, LGM400 provides an estimate of the LGM ocean that is similar to the modern one, except for an AABW that is much smaller in its extent.

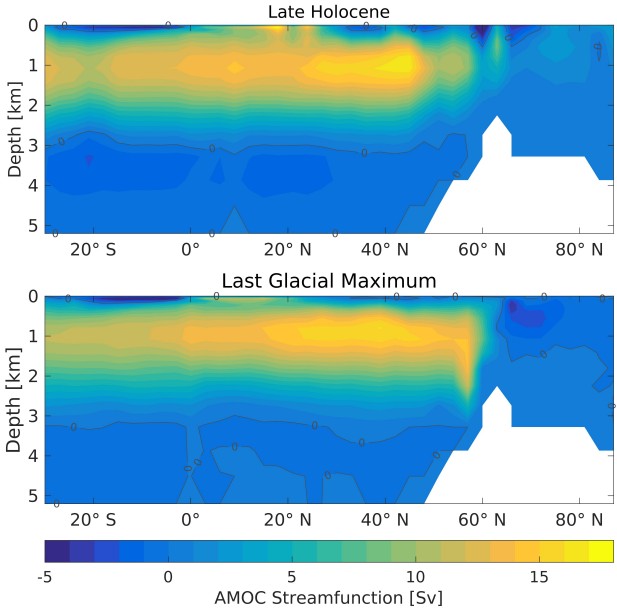

**Figure 5.** Simulated 100-year mean AMOC streamfunction in the LH estimate of Breitkreuz et al. (2018) and in LGM400.

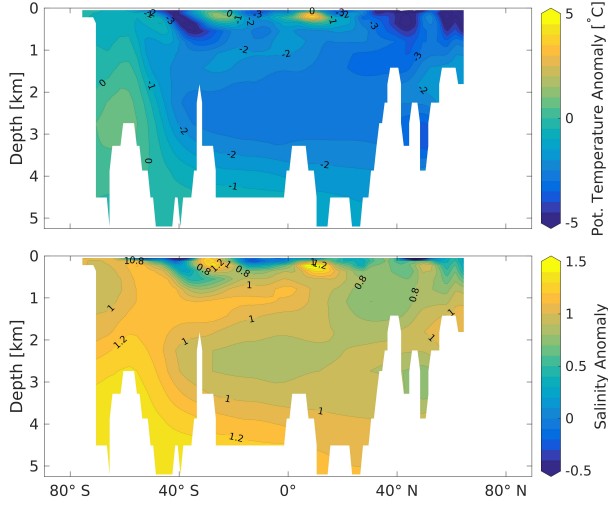

**Figure 6.** Potential temperature (upper panel) and salinity (lower panel) LGM400-LH anomalies. To compute the LGM-LH anomalies, we used the LH estimate of Breitkreuz et al. (2018).

## 3.3 Adjoint AMOC Sensitivities

The adjoint sensitivities in Figs. 8 and 9 provide information on how sensitive the AMOC is to changes in potential temperature and salinity around the global ocean. The patterns for temperature and salinity are very similar. Temperature and salinity



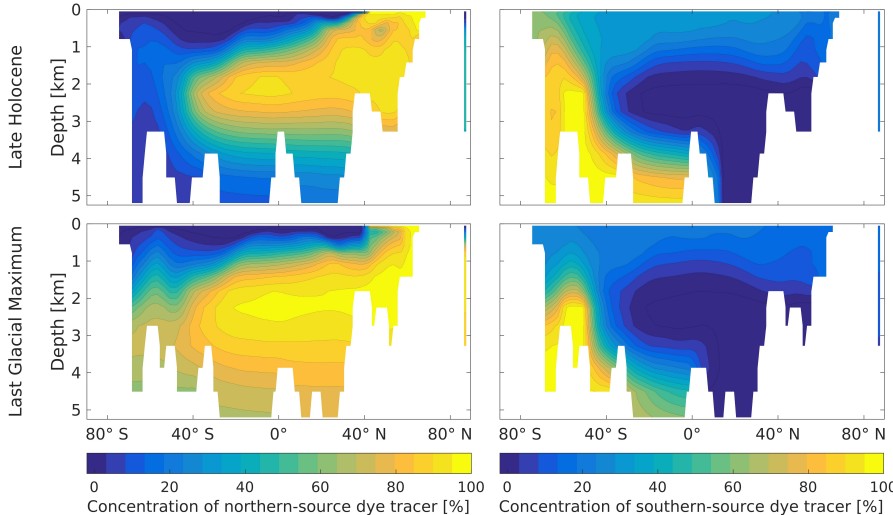

**Figure 7.** 100-year mean concentration of northern-source (left side) and southern-source (right side) dye tracer at a vertical transect through the Atlantic Ocean at 28° W in LGM400 and the LH estimate by Breitkreuz et al. (2018) at the end of a 2,000-year spin-up.

changes at the surface have the biggest impact in the North Atlantic and the Arctic Ocean (first row of Figs. 8 and 9), but their impact is small compared to changes in the deep ocean. One year before calculating the AMOC strength temperatures in the deep North Atlantic Ocean are most important. While the North Atlantic region still shows high sensitivities 150 years before calculating the AMOC, the deep Southern Ocean shows very high sensitivities as well. They are most pronounced in the

5 Atlantic and the Indian part of the Southern Ocean and reach up to about 30° S into the Atlantic and Indian Ocean. Sensitivities have decreased to smaller values after 400 years of the adjoint simulation but still show highest sensitivities in the Arctic Ocean, the North Atlantic and the most parts of the Southern Ocean.

## 4 Discussion

### 4.1 LGM Ocean State Estimate

According to a number of previous studies (e.g., Duplessy et al., 1988; Curry and Oppo, 2005; Lynch-Stieglitz et al., 2007; Muglia et al., 2018) the deep Atlantic Ocean was filled with a southern-source deep water mass overlaid by a shallow NADW during the LGM. With the adjoint method, we successfully found an LGM ocean state that is consistent with the employed data as well as with the physics of our model. However, in contrast to previous studies, our estimate shows a deep NADW and only a very weak extension of the southern-source AABW.

The modern Atlantic Ocean is dominated by a relatively salty NADW with waters originating from the tropics. The very deep southern Atlantic Ocean is filled with the fresher but colder AABW. For the LGM Atlantic Ocean it has been proposed that the AABW was much saltier compared to today due to increased sea ice formation and brine rejection, and that the glacial

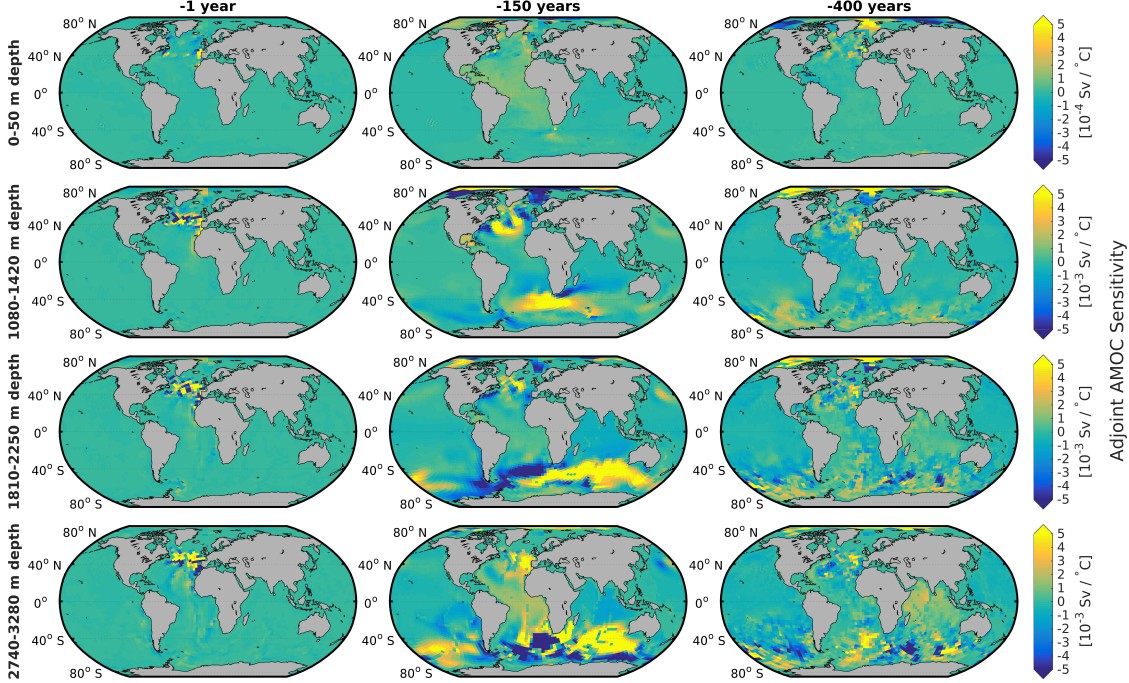

**Figure 8.** Adjoint sensitivities of the AMOC with respect to potential temperature 1, 150, and 400 years before evaluating the AMOC strength at the surface (0–50 m) and at depths of 1,080–1,420 m, 1,810–2,250 m, and 2,740–3,280 m. Note the change in scaling in the colorbar axis from the surface to deeper levels.

NADW was likely less salty relative to the glacial AABW (Adkins et al., 2002). Recently, however, a critical assessment of these results showed that while the proposed ocean state was possible, it was not required by the data (Miller et al., 2015; Wunsch, 2016). The global deep ocean is thought to have been overall substantially colder than the modern ocean (Adkins et al., 2002).

5     In our estimate these expectations are only partly met. The AABW is slightly saltier relative to the NADW (Fig. 6) and the deep ocean is mostly cooler, however, not as much as previously indicated (Adkins et al., 2002). Additionally, the negative temperature anomaly of the NADW is about 2 °C larger compared to the AABW (Fig. 6). Relative higher salinity and colder temperatures make the density change of the NADW in our estimate larger compared to the density change in the AABW. This coincides with the surface density changes in our LGM estimate, which show higher positive values for the region south

10 of Iceland where deep water is formed in our LGM estimate, but a smaller increase in density in the Southern Ocean in the Weddell Sea (Fig. 10). These density changes are, for example, in contrast to the results of Paul and Schäfer-Neth (2003) who found a stronger increase of density in the Weddell Sea compared to the North Atlantic and an AABW extending further north in their LGM simulation compared to their present-day simulation. The density changes might explain why the extend of the





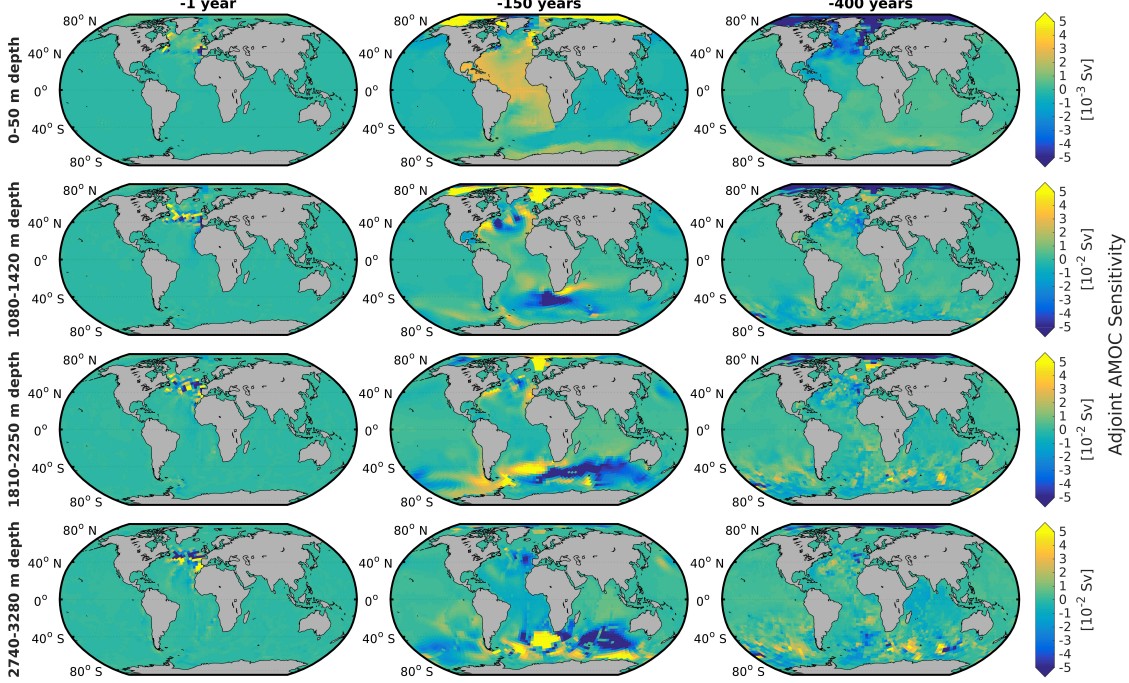

**Figure 9.** Adjoint sensitivities of the AMOC with respect to salinity 1, 150, and 400 years before evaluating the AMOC strength at the surface (0–50 m) and at depths of 1,080–1,420 m, 1,810–2,250 m, and 2,740–3,280 m. Note the change in scaling in the colorbar axis from the surface to deeper levels.

AABW into the Atlantic Ocean is very limited in our estimate. Additionally, the week counter-clockwise turning AMOC cell might contribute to the small northward extent of the AABW.

The AMOC in our estimate shows a very similar strength and depth compared with the modern ocean (Ganachaud, 2003; Breitkreuz et al., 2018). Previous studies using models (Otto-Bliesner et al., 2007; Weber et al., 2007; Muglia and Schmittner, 2015), a combination of models and proxy data via data assimilation (Kurahashi-Nakamura et al., 2017; Breitkreuz et al., 2019, in review), or $^{231}$Pa /$^{230}$Th data (McManus et al., 2004; Gherardi et al., 2005; Lippold et al., 2016) targeting the LGM AMOC strength have shown dissimilar results. Most recent studies inferred a weaker AMOC from the best fit of a physical-biogeochemical ocean model to $\delta^{13}$C, radiocarbon, and $\delta^{15}$N data (Muglia et al., 2018) and from $^{231}$Pa /$^{230}$Th and neodymium isotopes (Lippold et al., 2016; Howe et al., 2016).

## 4.2 Data Constraint

The estimated LGM ocean state indicates that the SST and the oxygen isotopic data alone do not necessarily support the presence of a shallower NADW and a weaker and shallower AMOC during the LGM. The model agrees well with the benthic $\delta^{18}$O$_c$ anomaly data below a depth of approximately 2,000 m. Only single data points indicate a mismatch and would require





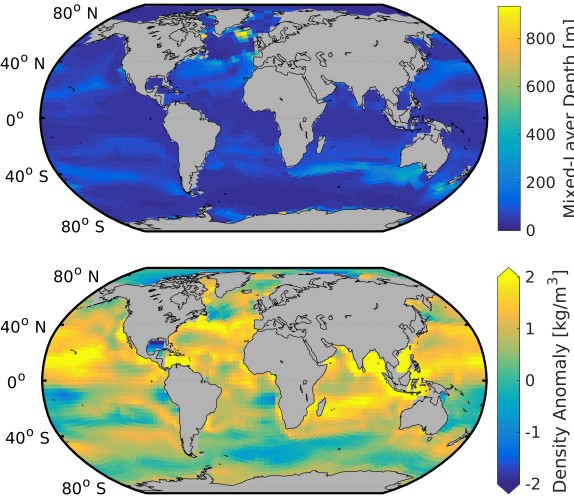

**Figure 10.** Mixed-layer depth in LGM400 (upper panel) and surface density anomaly LGM400-LH (lower panel). To compute the LGM-LH anomalies, we used the LH estimate of Breitkreuz et al. (2018).

higher anomaly values in the very deep Atlantic Ocean (Fig. 4). Higher (more positive) LGM-LH anomalies in the deep Atlantic Ocean would correspond to higher LGM $\delta^{18}O_c$ values, which in turn would correspond to higher $\delta^{18}O_{sw}$ values or lower temperatures in the deep Atlantic Ocean. Higher $\delta^{18}O_{sw}$ values could be reached if the NADW extended even deeper into the Atlantic Ocean, or in contrast, higher $\delta^{18}O_c$ values could be reached if the AABW was colder and penetrated further

into the deep Atlantic Ocean. The second option corresponds to the results of previous studies that indicate a larger extent of the AABW and colder temperatures close to the freezing point in the deep Atlantic Ocean (e.g., Adkins et al., 2002; Curry and Oppo, 2005).

  Above approximately 2,000 m the benthic $\delta^{18}O_c$ anomalies show some remaining model-data misfit (Fig. 4). The data would require lower LGM $\delta^{18}O_c$ values, which could be reached by lower $\delta^{18}O_{sw}$ values or higher temperatures. In the

southern part lower $\delta^{18}O_{sw}$ values could be reached by a stronger extension of the AAIW with a more depleted $\delta^{18}O_{sw}$ signal. A shallower NADW, however, would bring even more enriched $\delta^{18}O_{sw}$ values into this region and would then require much higher temperatures to meet the data.

  The issue that $\delta^{18}O_c$ is influenced by temperature and $\delta^{18}O_{sw}$ and that the effects of both tracers might cancel each other out has been previously acknowledged (e.g., Paul et al., 1999) and exacerbates the constraint of the circulation through only

$\delta^{18}O_c$ data. A likely reason for a very salty LGM AABW is increased sea ice formation (Adkins et al., 2002), but again, $\delta^{18}O_c$ data is not a good constraint for these salinity changes because $\delta^{18}O_{sw}$ is not as much influenced by sea ice formation and brine rejection as salinity is. Salinity estimates derived from pore-water measurements would be crucial in this regard but are still sparse and have high uncertainties (Wunsch, 2016).



Similarly, the AMOC strength seems to be not well constrained in our estimate. During the carry-over process the estimated AMOC strength varies between $11.0\,\mathrm{Sv}$ and $21.3\,\mathrm{Sv}$ (Table 2), which indicates that the estimated strength strongly depends on the length of the optimized simulation. A shorter simulation time might cause a less extensive transport of surface signals to the deep ocean and require a different circulation strength to reach the same state. However, the AMOC strength does not

decrease with the simulation length, which does not support this simple explanation. The AMOC strength in our final 400-year estimate is $16.1\,\mathrm{Sv}$ with the southward transport reaching a depth of $3{,}100\,\mathrm{m}$. The LGM estimate of Kurahashi-Nakamura et al. (2017), who used the adjoint method and the MITgcm in a similar configuration, yielded a stronger but shallower AMOC with a strength of $21.3\,\mathrm{Sv}$ and the southward transport reaching about $2{,}500\,\mathrm{m}$ deep, and a shallower NADW. The differences in the two estimates might be due to different observational data, slightly different model versions, or different choices of values

for model parameters. Kurahashi-Nakamura et al. (2017) additionally used benthic carbon isotopic data, whose constraint is missing in our estimate. However, they did not use any planktic isotopic data, such that the simulated surface values for $\delta^{13}\mathrm{C}$ and $\delta^{18}\mathrm{O}_{\mathrm{c}}$ were not constrained by proxy data. Additionally, the benthic data they used were confined to the Atlantic Ocean. Their optimized control variables were also used with a similar model configuration to obtain an equilibrium LGM simulation that showed a shallower and weaker AMOC, again highlighting the influence of the simulation length or of slight changes in

the model configuration on the result (Völpel et al., 2018).

Carbon isotopic data has shown to place a stronger constraint on the ocean circulation than oxygen isotopic data in previous studies (e.g., Marchal and Curry, 2008). The carbon isotopic composition in the deep ocean, however, depends on the remineralization of organic carbon. The implementation of carbon isotopes into the MITgcm, including $^{14}\mathrm{C}$, will be an important next step and allow to assimilate $\delta^{13}\mathrm{C}$ and $^{14}\mathrm{C}$ data, which will likely place a better constraint on the LGM ocean circulation.

Breitkreuz et al. (2019, in review) obtained an LGM ocean state estimate from the same observational data and the same model version, but their approach was based on a state reduction approach for the control variables and a Kalman smoother method. Their estimate shows a shallower NADW and a weaker AMOC with a maximum strength of $13.8\,\mathrm{Sv}$. However, in their estimate a substantially bigger model-data misfit remained and they found a second ocean state with similar model-data misfit but without an active overturning circulation in the Atlantic Ocean. The differences in the estimates suggest that results from

ocean state estimation highly depend on the assimilation approach, the choice of control variables, the remaining model-data misfit, and/or on the simulation length. While the estimate described here and that of Kurahashi-Nakamura et al. (2017) are limited to 400 years, the optimized simulations of Breitkreuz et al. (2019, in review) are 2,000-year equilibrium simulations. Additionally, the adjoint method as well as the Kalman smoother method used by Breitkreuz et al. (2019, in review) are highly first-guess dependent.

## 4.3   Adjoint AMOC Sensitivities

Additional to the type of proxy data, the location of the proxy data can be essential for the success in constraining the ocean circulation. Figures 8 and 9 indicate how sensitive the AMOC is to changes in potential temperature and salinity in different regions of the global ocean and, therefore, provide information on which areas are most important to be constrained by proxy data. The adjoint sensitivities for $\delta^{18}\mathrm{O}_{\mathrm{sw}}$ do not hold any information because $\delta^{18}\mathrm{O}_{\mathrm{sw}}$ is not an active tracer, that is, a change in



$\delta^{18}O_{sw}$ does not have an influence on the circulation or the AMOC. Nevertheless, the sensitivities to temperature and salinity changes indicate in which regions a constraint through different proxy data has the biggest effect.

Most important seem to be data in the North Atlantic Ocean and in the deep Southern Ocean. The North Atlantic region, where deep water is formed due to the density gradient between surface and deep water, is crucial for the strength of the

AMOC and hence, high sensitivities are visible for the surface as well as for the deep ocean in this region. While there is a good coverage of SST reconstructions for the North Atlantic (Fig. 1), the coverage of planktic and benthic isotopic data is not as extensive and more isotopic data from this region might place a better constraint on the simulated density gradient and improve future ocean state estimates.

The second region with highest sensitivities is the Southern Ocean. The Southern Ocean links all three major oceans via

the Antarctic Circumpolar Current (ACC). The density of the water in the Southern Ocean in the Atlantic region is, therefore, influenced by the southern regions in the Pacific and Indian Ocean via the ACC. In turn, the density of the South Atlantic Ocean is important in driving the Atlantic overturning circulation. These linkages might explain the high sensitivities of the AMOC to changes in the global Southern Ocean. Data coverage in the deep Southern Ocean is sparse in general and also in the data set used in this study. While some surface temperature estimates are available for the Southern Ocean, there is no benthic $\delta^{18}O_c$

data available south of approximately $50°$ S (Fig. 1).

Regarding the surface ocean, a constraint of the Atlantic Ocean seems to be more important than of the Indian or Pacific Oceans, supporting the results of Breitkreuz et al. (2019, in review). Whereas they could not determine the importance of deep-ocean data from outside the Atlantic Ocean from their results, the adjoint sensitivities specifically highlight the need for more benthic data in the southern part of the global ocean.

In general, sensitivities are highest 150 years before the AMOC is computed. In this and previous LGM ocean state estimates the cost function covered only the last 5 years (Dail and Wunsch, 2014), 10 years (Kurahashi-Nakamura et al., 2017), 60 years (this study), or 100 years (Breitkreuz et al., 2019, in review) of the optimized simulation. The adjoint sensitivities indicate that an extension of the cost function to a longer time interval might help in obtaining better constrained ocean state estimates. An extension of the cost function in an adjoint simulation, however, creates high recomputation and storage costs because the

required state variables in each time step during the cost function interval are needed for the adjoint calculation.

## 5    Conclusions

We presented a new ocean state estimate that was obtained from an oxygen isotope-enabled general circulation model and a successful application of the adjoint method. The adjoint method provides a solution that is consistent with the employed data and the model physics. The estimate extends previous state estimates by using benthic as well as planktic data on the oxygen

isotopic composition of calcite. It is additionally constrained by global annual and seasonal SST reconstructions. It shows an LGM Atlantic Ocean that is dominated by a northern-source deep water mass corresponding to the modern NADW. The deep southern-source water mass is less extensive than during the LH ocean, in contrast to previous studies. The unexpected results might be due to an insufficient constraint of the available oxygen isotopic and SST data. Comparisons with previous ocean state





estimates indicate that results from ocean state estimation strongly depend on the assimilated proxy data and the experimental design. A better coverage of proxy data of different types could help to obtain more reliable ocean state estimates. Adjoint sensitivity experiments revealed that especially benthic data from the North Atlantic and from the global Southern Ocean are most important for successfully constraining the AMOC strength.

5 *Code and data availability.* The MITgcm model code is publicly available via http://mitgcm.org/. The oxygen isotope data set used in this study will be submitted to PANGAEA (https://www.pangaea.de/) by the authors of Breitkreuz et al. (2019, in review).

*Author contributions.* A. Paul and M. Schulz were involved in the funding acquisition and supervision of the project. A. Paul, M. Schulz, and C. Breitkreuz designed the experiments and C. Breitkreuz carried them out and analyzed the results. C. Breitkreuz created all figures and tables in the manuscript and wrote the manuscript. All coauthors reviewed and commented on the manuscript.

10 *Competing interests.* The authors declare that they have no conflict of interest.

*Acknowledgements.* This study was funded by the German Federal Ministry of Education and Research (BMBF) as a Research for Sustainability initiative (FONA) through the German Climate Modeling Initiative PalMod (FKZ: 01LP1511D). We are grateful to Rike Völpel for providing the water isotope module for the MITgcm.





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
