# Peer review of "A dynamical reconstruction of the Last Glacial Maximum ocean state constrained by global oxygen isotope data"

_Climate of the Past, 2019_

## Referee Comment (RC1) · Anonymous Referee #1 · 30 May 2019

This paper is an interesting and potentially significant one. Many of the comments that follow are directed at amplification, explanation, and better definitions. In a field such as paleo-modelling, one is dealing with a very diverse audience and it becomes even more important than normal to make sure that readers will not be misled or misunderstand.

Many papers modelling the LGM and other periods have taken pains to demonstrate that their model does at least a reasonable job of describing the modern system. Readers would benefit from knowing e.g., what the current set up does with the modern AMOC and other features such as the heat transport, depth of mixed-layer, etc. (credibility of the results are at stake). There is a brief reference to an earlier paper directed

at oxygen distributions, but one should not have to hunt that down in order to have a sense of what the model capability is or isn't.

The AMOC is the center-piece of the paper, but it is never even defined here! Perusal of the modern literature shows a variety of definitions, involving depths of integration (different density surfaces, physical depths, depth of maxima defined somehow, latitudes, etc.). Values differ by an order of magnitude by latitude and sometimes even depth and integration intervals and show little or no covariance with latitude, a result going back at least to Bingham et al. GRL 2007 and several subsequent studies using data.

The Gulf Stream is the dominant component of the AMOC (see especially some of the recent ECCO discussions of AMOC and/or the RAPID array data). What is the strength of the western boundary currents? No paleo-scientist, unfamiliar with the modern literature, would appreciate how time and space variable the modern AMOC is, however it is defined. The authors do use 45N latitude, but with no discussion of how that cuts across the Gulf Stream system in the modern ocean, rendering it particularly spatially and temporally noisy. They say "southward transport"–but to what depth, over what zonal integral? Modern average results show a complex of both northward and southward flows, changing with depth with boundary currents on both coasts.

No modeler would ever try to construct the modern circulation without a considerable knowledge of the wind system. Winds are barely even mentioned beyond the statement that they are taken from a model, and in passing, as part of the control vector. Given the dominance of wind forcing, particularly its curl, in the modern circulation, the reader needs some discussion of sensitivity to it and/or the reliability of the field that was used/emerged.

"Kalman smoother" is not standard terminology and tends to suggest the authors are not fully aware of the optimization literature. There is a Kalman filter (a predictor) and there are lots of smoothers (RTS, fixed lag, fixed interval, etc.) Any optimization

textbook gives definitions. What is intended?

The discussion of SST, the isotopic tracers, and MARGO on P. 4-6 left me totally confused. Are the oxygen isotope tracers inconsistent with MARGO? In light of the misfits found later, one should know a priori. MARGO is based on proxies as well, and labelling it as SST as opposed to planktic and benthic isotopes is a strange distinction.

Table 3. If I'm interpreting this correctly (?), some of the residuals look far to large (e.g., S0, precip., et al., and some look much too small e.g., F8. Unless I'm misunderstanding (?), such results would lead to formal rejection of a solution.

Table 5. The imposition of hard maxima and minima is not a part of conventional least-squares. How were these implemented?

P. 3 A better discussion of the ECCO model adjoint is Heimbach, P., Hill, C., Giering, R., 2005. An efficient exact adjoint of the parallel MIT general circulation model, generated via automatic differentiation. Future Gener. Comput. Syst. 21 (8), 1356–1371.

P. 14, line 14. Should be 'decrease'

P. 11 and elsewhere. Given the prominence of MARGO data in previous estimates, it would be helpful to know why the MARGO cooling is not seen here? Is this a systematic misfit to MARGO data? How do the N. Atlantic results compare e.g., to Dail's?

P. 16, line 4. The objective function is a conventional quadratic misfit. How is it possible for it to become non-convex? Maybe owing to the presence of the Lagrange multipliers in it (although that is never mentioned)?

Table 2 and elsewhere. Are the underlying variables at least approximately Gaussian? That's required for the chi-square fit to be used.

The re-weighting of the control portion of the objective function in the equation on P. 7 implies a major change in the errors relative to Jdata. Does that make physical sense?

Section 2.4 seems pointless: there is no inference or conclusion.

P. 14. Worth explaining that in the modern ocean there exists a theory of AMOC sensitivity to Southern Ocean winds (Toggweiler).

P. 14, line 15. Isn't the N. Atlantic regarded as relatively salty mainly because of the Med. Water outflow?

P. 15 line 5. Is it possible there is a systematic misfit to the data in the NADW and deep ocean? Or is that ruled out?

P. 16. The meaning of the phrase "do not necessarily support" is not obvious to me. Not consistent with?

P. 17 "exacerbates" is a peculiar adjective. Maybe "exaggerates"?

P. 18. Is the flow steady at 16.1 Sv or is there time variability? And what is the heat transport of the estimate? That's an essential element of the climate state. Is a North Atlantic state without any overturning physically feasible? How is the heat budget balanced? Is it geostrophic? line 31 "Additional to the type of.." isn't English.

P. 19 The second sentence, claiming consistency with all the data appears to contradict much of what has come before. And "consistency" here, given the vagueness of the error discussion is not easy to interpret. MARGO?

It would be helpful to have a broader discussion of the most important of the consistencies or otherwise, of the previously published solutions including Kurahashi-Nakamura et al., but also Amrhein, Thornalley, Gebbie, Oppo, etc.

---

## Referee Comment (RC2) · Anonymous Referee #2 · 13 Jun 2019

The authors used the coupled ocean-sea-ice MIT general circulation model to estimate the LGM circulation state based on anomalies of reconstructed SSTs and foraminiferal oxygen isotopes. The approach is innovative and supports the growing evidence that the LGM circulation state was not necessarily associated with a shallower North Atlantic Deep Water and stronger Antarctic Bottom Water. Below are mostly minor recommendations that the authors may want to consider:

1. P4/L3, P6/L28: d18O anomalies are used in order to eliminate species specific vital effect. Generally, the use of anomalies is a good idea, because it minimizes constant laboratory offsets or systematic habitat effects, i.e. if seasonal or vertical

temperature changes are correlated. However, it is important to keep in mind that not all vital effects are constant, but might also change with environmental conditions. For example, changes in pH or light conditions can affect the d18O composition of foraminiferal shells. I find the use of anomalies reasonable, but suggest that the authors adapt their justification accordingly.

2. It is not entirely clear to me how the LH-LGM d18O anomalies of foraminifera have been exposed to the model. Have the differences between LH and LGM been added to the run of Breitkreuz et al. (2018) or are the anomalies the raw differences derived from the data? What is the exact time slice for the late Holocene, the last 2 kyrs or the last 4 kyrs? Please expand.

3. P3/L19/20/Table 1: Note that not all d18O data from Völpel et al. (2019) are of late Holocene age. The top of GeoB9510-1 is at least of mid-Holocene age (see Figure 2 & 3 in Völpel et al., 2019). Please clarify.

4. P2/L3: The authors write "Many studies indicate the presence of a shallower NADW and a more sluggish AMOC during the LGM compared to today (Lynch-Stieglitz et al., 2007". Actually, Lynch-Stieglitz et al. are quite vague on this point and mention evidence for both interpretations, a deeper and a shallower NADW. For example, they write "...this finding suggests that waters originating in the North Atlantic also contributed to the deep (>2km) water mass in the LGM Atlantic. This also argues against a much slower circulation for both deep-water masses in the Atlantic.". I suggest to cite specific papers for both scenarios.

5. P16/L13: "Only single data points show a mismatch in areas where other data points indicate agreement (Fig. 2)." Can the authors be more specific on potential reasons why these data points are different? Is it a specific proxy that does not work? Are there differences in sedimentation rate?

6. It would be interesting to know, to what extent the modelled LGM surface water d18Ow is consistent with reconstructions. For example, Duplessy et al. (1991,

Oceanologica Acta 14, 311-324) have reconstructed water with high d18O/salinity near 30-40°W, north of the polar front.

7. P13/14/17: The authors compare the LGM state with a modern state from Breitkreuz et al. (2018), which they denote "Late Holocene" (LH) estimate. Is this strictly correct and the experiment truly representative for the late Holocene, considering that the Breitkreuz et al. (2018) runs have been constrained with modern (past 1950) oceanographic data? These data might already contain a global warming signal.

Minor points

-P1/L8: (check in entire text): "benthic as well as planktic data on the oxygen isotopic composition of calcite", better "oxygen isotope ratio of benthic and planktic foraminifera"

-P1/L15: Cite Mix et al. (2001) for the definition of the LGM time slice

-P3/L10: "Massachussets" must be "Massachusetts"

-P4/Table 1/References: "Völpel et al. (2018)" must be "Völpel et al. (2019)"

-P17/L14: Also cite Zahn & Mix (1991, doi:10.1029/90PA01882)

---

## Referee Comment (RC3) · Anonymous Referee #3 · 18 Jun 2019

Summary

This paper applies the adjoint gradient method in an ocean general circulation model to the problem of ocean state estimation at the Last Glacial Maximum. The state estimate is constrained by upper-ocean temperature data and upper-ocean and benthic data for the oxygen isotope ratio of calcite. The inclusion of a new data type (planktonic foraminiferal oxygen isotopes) and new control variables (isotopic composition of precipitation and water vapor) make this paper a useful and novel contribution to the field. At this stage I am recommending major revisions because of some outstanding questions about the methodology and results, detailed further below.

[Figure]

Major points

1. A key result that the authors have omitted is what the changes to the model controls look like. Are they consistent with possible atmospheric conditions at the LGM? In addition to giving the reader a sanity check (particularly for the isotopic controls, which are novel in this work), plotting the control adjustments can provide physical insights into how and why the model is fitting the data. Given that this configuration also adjusted ocean mixing, it would be interesting to see how fields of mixing parameters were changed to fit the data.

2. The authors apply two different normalizations and multiplicative factors to the computation of the control component of the cost function. While the proof is ultimately in the pudding – a solution was found – these factors (the N_data/N_ctrl term and the "preconditioning") change the statistical interpretation of the cost function and the adjoint procedure. In particular, I suspect that the J'_ctrl term, which is an important indicator of whether the control adjustments have a reasonable amplitude relative to their a priori uncertainties, means something different under all of these scalings. I highly encourage the authors to simplify their description and put the various scalings into the control uncertainties, which is mathematically identical to what they have done already but more intuitive to the reader.

3. I wonder whether the experiments with adjoint sensitivities to AMOC would be better explored in more detail in a separate paper. Some of the challenges in making sense of sensitivities are illustrated by modern AMOC sensitivity studies of Pillar et al. 2016 and Kostov et al. 2019. Extending these sensitivities to decide which records are useful and which are not is potentially powerful, but should probably be approached with caution. Are these sensitivities large enough to be meaningful? Is the system linear enough that they apply under forcings similar to actual ocean variability? How can we use transient sensitivities to inform equilibrium sensitivity? How does this approach take into account that some regions are already relatively well sampled? See e.g. Comboul et al. 2014 for an application of sensor placement procedures to paleo problems.

4. It would be useful to give some more information about the number of adjoint itera-tions you used and your criterion for solution convergence.

5. How unique is this solution likely to be?

6. It would be useful to provide more comparisons of the solution with other state estimates and to rationalize their differences.

Line-by-line comments

p1l9 Please give an age range for Late Holocene.

p1l12-13 This statement could use clarification. What kinds of data? On what time scales?

p1l16 today's

p1l21 extent, not extend, though you could consider being more specific

p1l18 Suggest "Some estimates are based only on... while others include the assimilations..."

p1l21 It is not clear what "longest" means in this context

p3l6 Suggest "an indication of which... are most important to measure in order to constrain AMOC strength."

p4l2 Time anomalies

p3l18 Please define time ranges you are targeting for LGM and LH

p4l11 and elsewhere: I am not sure what CoP's policy is on citing work that has not been published. It would be helpful to see the manuscript of Breitkreuz et al. 2019.

p5l9 What is the origin of the first-guess controls?

p5l13-14 Please clarify this statement ("the longer the simulation, the more difficult...") Is it that it is difficult to reduce the cost function quickly? It would be helpful to provide

an example or citation. I don't know that the Evensen paper deals with long time scales or adjoint methods.

p5l19 Is there a contribution to d18Ow due to evaporation as well?

p6l3 By the isotopic control variables, do you mean the isotopic composition in precipitation and water vapor? It's confusing why eliminating a control variable would make the fit worse.

p6l4 I don't think you have established that it is non-convex(?) so suggest "non-convexities" rather than "the non-convex shape"

p6l7 Large changes compared to what?

p6l11 Please state that these are the LGM-LH anomalies.

p7l13-14 Including such a scale factor changes the interpretation of the cost function and the role of the control uncertainties. Taking a step back, if you assume that there is a model-data misfit and that the data uncertainties are correctly specified, if one finds that the controls are not being adjusted, the suggestion is that the prior control uncertainties are too small. Multiplying by this factor has the effect of increasing these uncertainties (the sigma in Table 3) by a factor of $\sqrt{N\_ctrl/N\_data}$.

I think it is useful and important to report the scaling this way to the reader (as inflated uncertainties) because it is a key control on how much the model can be changed to fit the data. In particular, J'_ctrl in Table 3 and elsewhere should be adjusted to the unscaled case, lest the impression be given that very small control changes sufficed to fit the data.

p8l5 "supports..." Can you clarify? The impression seems to be that including this term makes the solution equilibrated; is that the case?

p8Eq1 (mislabeled, not the first equation): Annual mean temperature where? surface ocean, all depths? What does it mean that there are drifts in annual mean sea surface

elevation – is sea level changing? Is the variance in space meant instead? Also, please clarify what is meant by southward transport – net?

p8l7 Please elaborate on what is meant by "a sufficient reduction of the cost function" and describe more how these weights are derived.

p8l8 Please elaborate on this preconditioning. "Size" could be replaced by the clearer "amplitude" if that is in fact the meaning. Is this normalization in addition to that coming from the multiplication by the Ws in the equation on page 7? If so, this additional preconditioning should also probably be included in the computation of J'_ctrl in Table 2.

p9l10 Please clarify why the model was run out in this repeating fashion. Was it a computational convenience? Does the model drift strongly?

p9l16 Strength at which depth level?

p10l6 A value below one can also point to the undesirable result of overfitting the observations. Is there any danger of that happening here with the value of J_misfit'=0.7 for SSTs? The posterior misfits appear to be quite small, though it's hard to tell because they have not been normalized by their uncertainties.

Figures 2 and 3: I would consider not whiting out misfits that are lower than uncertainty. It makes it look like the data are being overfit.

p10l7-8 "Only single. . ." I'm not sure what you meant by this.

p10l9 And in the North Atlantic subpolar gyre!

p10l10 I would avoid "greatly" and "small" or put in some values for comparison.

p10l14 Can you say more about what is meant by "locally big" and "implausible"?

p11l1 I don't think J'_ctrl represents global mean changes in individual control fields; rather, it's the sum of squared, /local/ changes normalized by uncertainties

p11l4 Why is global mean sea surface elevation drifting – is there an imbalance in E-P-R? Also, is this temperature drift measure for the upper ocean? The entire ocean? It would be helpful to have some numbers to compare these to (e.g., modern interannual variability) so that the reader can assess the extent to which the model is driving.

p11l5 Can you be more specific than "the center of the AMOC"?

p11l17 Couldn't these high anomaly values also be due to the control adjustments?

Figure 4: Please say what the lower panel is.

p12l9 Please clarify what about the estimated AMOC is weaker

p12l12 Please clarify what is meant by "extent"

p12l11 I think there is room in this section to provide some more results and discussion if you would like to make the broad statement that the estimated LGM state ocean is similar to the modern (LH?). Or shift the focus of the statement to the Atlantic.

Section 3.3 It is important to emphasize that these are linear sensitivities. Also, what is the meaning of the sensitivities? Are these anomalies that are imposed for a certain time, at a certain lag? Please clarify in the text.

p13l1 Please restate what is meant by "the AMOC" – strength, structure, etc.

p14l1 This is a little confusing; it sounds like the Arctic and North Atlantic are affected by surface ocean changes.

p15l9-10 higher... smaller... Relative to LH? Please specify.

p15l14 extent, not extend

p16l1 This seems a bit chicken-and-egg to me.

p16l5 Can you discuss why the results might be dissimilar? Might it be due in part to the fact that the model is still spinning up?

p16l10 The meaning of this section title is not clear.

p16l11 I would say instead that they do not require the presence of a shallower NADW.

p16l13 What is meant by single data points indicating a mismatch? Please clarify.

Fig. 10 Is MLD ever referenced in the text?

p17l2 Why not lower LH values?

p17 l9-10 Please be more specific than "the southern part"

p18l1-2 But are the misfits the same?

p18l6 The "southward transport" seems to be used as a measure of the separation between upper and lower cells. Please spell this out more clearly.

4.3 These sensitivities are for various transient lag times. How do they inform the equilibrium sensitivity, which seems to be relevant for the LGM problem?

p18l20 It would be useful to have a comparison with other LGM state estimates (Dail, etc.) and a discussion of why the results might be different.

p19l1 I think the authors should exercise greater caution in interpreting the adjoint sensitivities as observational placement experiments. For one, they ignore the locations where we already have observations. Second, the are linear (perturbation) sensitivities. It would be worth testing (using a set of perturbations of various sizes) whether the response to these patterns remains linear at amplitudes that we might expect to see in nature. See the examples of Pillar et al. 2016 and Kostov et al. 2019.

p19l12-13 Please provide a citation for this claim

p19l16 Confusing; please rephrase.

p19l23 What is meant by the extension of the cost function... Longer adjoint simulations?